# Nonperturbative Kinetic Description of Electron-Hole Excitations in Graphene in a Time Dependent Electric Field of Arbitrary Polarization

**Stanislav A. Smolyansky** [1,2] , **Anatolii D. Panferov** [1] , **David B. Blaschke** [3,4,5,*] **and**
**Narine T. Gevorgyan** [6]

1   Saratov State University, 410026 Saratov, Russia; smol@sgu.ru (S.A.S.); panferovad@sgu.ru (A.D.P.)
2   Department of Physics, Tomsk State University, 634050 Tomsk, Russia
3   Institute of Theoretical Physics, University of Wrocław, 50-204 Wrocław, Poland; david.blaschke@gmail.com
4   Bogoliubov Laboratory for Theoretical Physics, JINR Dubna, 141980 Dubna, Russia
5   Peoples' Friendship University of Russia (RUDN University), 117198 Moscow, Russia
6   Russian-Armenian University, 0051 Yerevan, Armenia; gevorgyan.narine@gmail.com
*   Correspondence: david.blaschke@gmail.com; Tel.: +48-71-375-9252

**Abstract:** On the basis of the well-known kinetic description of $e^- e^+$ vacuum pair creation in strong electromagnetic fields in $D = 3 + 1$ QED we construct a nonperturbative kinetic approach to electron-hole excitations in graphene under the action of strong, time-dependent electric fields. We start from the simplest model of low-energy excitations around the Dirac points in the Brillouin zone. The corresponding kinetic equations are analyzed by nonperturbative analytical and numerical methods that allow to avoid difficulties characteristic for the perturbation theory. We consider different models for external fields acting in both, one and two dimensions. In the latter case we discuss the nonlinear interaction of the orthogonal currents in graphene which plays the role of an active nonlinear medium. In particular, this allows to govern the current in one direction by means of the electric field acting in the orthogonal direction. Investigating the polarization current we detected the existence of high frequency damped oscillations in a constant external electric field. When the electric field is abruptly turned off residual inertial oscillations of the polarization current are obtained. Further nonlinear effects are discussed.

**Keywords:** graphene; dynamic critical phenomena; high-field and nonlinear effects

**PACS:** 81.05.Uw, 64.60.Ht, 73.50.Fq

## 1. Introduction

In recent years considerable interest has developed in a nonperturbative, dynamical description of transport phenomena in condensed matter physics inspired by the physics of strong electromagnetic fields [1]. Particular attention was devoted to graphene (see, e.g., [2,3]). In this case there is an obvious similarity with the dynamical Schwinger effect in QED, the creation of electron-positron pairs from the vacuum in strong electromagnetic fields [4–6]. In this context the nonperturbative kinetic approach has proven successful. It is based on the transition to a quasiparticle representation in the presence of an external, quasiclassical electric field facilitated by a time dependent Bogoliubov transformation [7–10]. It would be natural to adopt these methods to specific problems in condensed matter physics and, in particular, to the physics of graphene. Such an adaptation is performed in the present work. The application of these methods allows for advancement to nonperturbative investigations of nonlinear effects in graphene in the presence of strong external electric fields.

We want to give a detailed outline of the contents of the present work. In Section 2, Section 2.1 the basic kinetic equation (KE) for the simplest dynamical model of graphene [2,3,11–14] (a single layer graphene sheet with two Dirac points of the Brillouin zone and absence of the standard scattering mechanism of carriers) is obtained using nonperturbative techniques for the case of a spatially homogeneous, time-dependent external electric field of arbitrary polarization in the graphene plane. The transition to the quasiparticle representation is obtained with the help of a unitary transformation expressed in explicit form [15]. All subsequent consideration is essentially nonperturbative.

The process of electron-hole (*e-h*) pair creation in a strong electric field can be considered as a specific field-induced phase transition in a system with broken symmetry [1,10,16]. In Section 2.2 some features of this process are considered in graphene. Section 3 is devoted to the connection of observables such as the quasiparticle number and current densities with the kinetic theory. Here we discuss also the energy conservation law for a system in an external electric field. Here, in particular, an order parameter is introduced which describes the polarization properties of graphene. It is shown that after switching off the external field pulse the order parameter survives and oscillates with momentum dependent amplitude. In other words, the evolution of the order parameter is defined by the entire prehistory of the graphene evolution during the application of the external field. In particular, this effect becomes apparent in the damped oscillations of the residual polarization current on the background of a constant residual conduction current (Section 4). Here it is also shown that the polarization current dominates over the conduction current. This dominance turns out also in calculations of the currents in the framework of the standard perturbation theory. For example, in the Appendix A, we reproduce the well-known results for the polarization and conduction currents in the leading orders of the expansion with respect to $E/E_0 \ll 1$, where $E_0$ is the characteristic field (1).

Section 5 contains results of the numerical calculations of the distribution functions of the carriers for electric fields of different magnitude and spectral composition models both for linear and elliptic polarizations. This fact is an important hint that the similar situation is valid also in $D = 3 + 1$ QED, where analogous calculations can be very complicated [17].

In Section 6 we outline the effect of manipulating a weak signal with a current by means of generating active properties of graphene with the help of another (basic) field.

Finally, in Section 7, by analogy with Section 2, we derive the KE in the $D = 2 + 1$ tight binding model of the nearest neighbor interaction [3,11,12]. Also in this case the conduction and polarization currents are obtained. Their detailed investigation will be performed in a separate work.

The conclusions are drawn in Section 8.

We use the metric $g_{\mu\nu} = \text{diag}\,(1, -1, -1)$ and the coordinates $x^{\mu} = (v_F t, x^1, x^2)$. We will proceed from the basic parameters of the model: $a = 2.46$ Å is the lattice spacing, $\gamma = 2.7$ eV is the hopping energy, and $v_F = 10^6$ m/s is the Fermi velocity. We define a set of scale factors for the physical quantities time ($t_0$), momentum ($p_0$), and field strength ($E_0$) according to

$$t_0 = \frac{a}{v_F}, \quad p_0 = \frac{\hbar}{a}, \quad E_0 = \frac{\hbar v_F}{ea^2}. \tag{1}$$

## 2. Kinetic Equation

In this section the basic KE for the description of electron-hole excitations in external, time-dependent electric fields will be derived for the $D = 2 + 1$ QED model of graphene in the framework of a low-energy model (for a tight-binding model, see Section 7 below). Some necessary prerequisites for such a derivation have already been obtained earlier [15] by means of the diagonalization of the initial Hamiltonian of the model. Our approach is based on the consistent usage of the occupation number representation and the adaptation of a method that is well known in $D = 3 + 1$ QED for the description of the creation of an electron-positron plasma from the vacuum in strong fields [7,8,18].

Let us assume the graphene layer is located in the plane $(x^1 = x, x^2 = y)$. A time dependent spatially homogeneous electric field acts in this plane, i.e., the corresponding vector potential in the Hamiltonian gauge is $A^k(t) = (0, A^1(t), A^2(t))$. The spatial homogeneity of the electric field can be provided, for example, in the focal spot of two coherent laser beams counter propagating along the axis perpendicular to the graphene layer. It is assumed that the field model is finite, i.e., that the field strength $\vec{E}(t) = -\frac{1}{c}\dot{\vec{A}}(t)$ vanishes before switching on and after switching off the laser fields, $\lim_{t\to\pm\infty} E(t) = 0$ (the dot above the symbol denotes its time derivative). This is necessary for the correct definition of the in- and out- states of the vacuum with $A_{\text{in}} = A(t \to -\infty)$ and $A_{\text{out}} = A(t \to \infty)$.

### 2.1. The Low-Energy Approximation

The Dirac-type equation for the low-energy excitations in graphene in a time dependent electric field described above is

$$i\hbar\dot{\Psi}(\vec{x}, t) = v_F \hat{\vec{P}}\vec{\sigma}\Psi(\vec{x}, t),\tag{2}$$

where $\hat{P}_k = -i\hbar\nabla_k - (e/c)A_k(t)$ is the quasi-momentum ($k = 1, 2$) and $\sigma_k$ are the Pauli matrices corresponding to the pseudospin structure of graphene.

The Hamiltonian of the theory,

$$H(t) = \frac{i\hbar}{2}\int d^2x\left[\Psi^\dagger(\vec{x}, t)\dot{\Psi}(\vec{x}, t) - \dot{\Psi}^\dagger(\vec{x}, t)\Psi(\vec{x}, t)\right],\tag{3}$$

is the 00 component of the corresponding energy-momentum tensor and it can be transformed with help of the equation of motion (2) to the form

$$H(t) = v_F\int d^2x\Psi^\dagger(\vec{x}, t)\hat{\vec{P}}\vec{\sigma}\Psi(\vec{x}, t).\tag{4}$$

Here we dropped the spin indices.

The wave function here is a two-component spinor permitting the decomposition

$$\Psi^T(\vec{x}, t) = \frac{1}{(2\pi\hbar)^2}\int d^2p\left(\Psi^{(1)}_{\vec{p}}(t), \Psi^{(2)}_{-\vec{p}}(t)\right)e^{i\vec{p}\vec{x}/\hbar},\tag{5}$$

which translates the Hamiltonian function (4) to the momentum representation.

For the physical interpretation of the model it is appropriate to go over to the quasiparticle representation, where the Hamiltonian of the theory is diagonal. As it was shown in the work [15], this is achieved with the unitary transformation

$$U^\dagger(t)v_F\vec{P}\vec{\sigma}U(t) = \varepsilon(\vec{p}, t)\sigma_3 = H_{\vec{p}}(t),\tag{6}$$

and $\Phi = U^\dagger\Psi$ with the unitary matrix [15]

$$U(t) = \frac{1}{\sqrt{2}}\begin{pmatrix}\exp(-i\varkappa/2) & \exp(-i\varkappa/2) \\ \exp(i\varkappa/2) & -\exp(i\varkappa/2)\end{pmatrix}.\tag{7}$$

The function $\varkappa$ is defined by the condition (6) [15], corresponding to $\tan\varkappa = P^2/P^1$, where $P^k = p^k - (e/c)A^k(t)$. The quasienergy $\varepsilon(\vec{p}, t)$ in (6) is determined by the dispersion relation in the vicinity of the Dirac points

$$\varepsilon(\vec{p}, t) = v_F\sqrt{P^2} = v_F\sqrt{(P^1)^2 + (P^2)^2}.\tag{8}$$

Equation (2) transforms then to the form

$$i\hbar\dot{\Phi} = H_{\vec{p}}(t)\Phi + \frac{1}{2}\lambda\hbar\sigma_1\Phi,\tag{9}$$

where $H_{\vec{p}}(t)$ is defined by Equation (6) and

$$\lambda\,(\vec{p},t) = \dot{\varkappa} = \frac{ev_F^2[E_1P_2 - E_2P_1]}{\varepsilon^2(\vec{p},t)}. \tag{10}$$

Introducing the notation

$$\Phi(\vec{p},t) = \begin{bmatrix} a(\vec{p},t) \\ b^\dagger(-\vec{p},t) \end{bmatrix}, \tag{11}$$

the Hamiltonian function (4) can be rewritten in the quasiparticle form

$$
\begin{aligned}
H(t) &= \int [dp]\varepsilon(\vec{p},t)\Phi^\dagger(\vec{p},t)\,\sigma_3\Phi\,(\vec{p},t) \\
&= \int [dp]\varepsilon(\vec{p},t)\left[a^\dagger(\vec{p},t)a(\vec{p},t) - b(-\vec{p},t)b^\dagger(-\vec{p},t)\right],
\end{aligned}
\tag{12}
$$

where the abbreviation $[dp] = d^2p(2\pi\hbar)^{-2}$ has been used.

Apparently, the realization of the unitary transformation in the explicit form in both the low-energy and the tight-binding (see below Section 7) models is a result of the fact that these models belong to the class of conformal-invariant field theories (see, e.g., Ref. [6]).

At this stage one can go over to the occupation number representation and replace the amplitudes $a^\dagger(t)$, $a(t)$ and $b^\dagger(t)$, $b(t)$ by the corresponding creation and annihilation operators for electrons and holes considered as quasiparticles. These operators are defined on the in-vacuum state $|\text{in}\rangle$ with vector potential $\vec{A}_{\text{in}}$ and satisfy the canonical anti-commutation relations

$$\left\{a(\vec{p},t), a^\dagger(\vec{p}',t)\right\}_+ = \left\{b(\vec{p},t), b^\dagger(\vec{p}',t)\right\}_+ = (2\pi)^2\delta(\vec{p} - \vec{p}'). \tag{13}$$

Other elementary anti-commutators are equal to zero.

From Equations (2), (6) and (11) it follows the equations of motion of the Heisenberg type for the description of the unitary evolution of the creation and annihilation operators, e.g.,

$$\dot{a}(\vec{p},t) = \frac{i}{\hbar}\,[H(t), a(\vec{p},t)] - \frac{i}{2}\lambda\,(\vec{p},t)\,b^+(-\vec{p},t) = \frac{i}{\hbar}\,[H_{tot}(t), a(\vec{p},t)], \tag{14}$$

$$\dot{b}(\vec{p},t) = \frac{i}{\hbar}\,[H(t), b(-\vec{p},t)] + \frac{i}{2}\lambda\,(\vec{p},t)\,a^+(\vec{p},t) = \frac{i}{\hbar}\,[H_{tot}(t), b(-\vec{p},t)], \tag{15}$$

where the amplitude of the transitions between states with the positive and negative energies of the quasiparticles is defined by Equation (10). From Equations (9), (14) and (15) it follows that evolution of the system is unitary. The Fock space is constructed on the time dependent vacuum state. In Equations (14) and (15) $H_{tot} = H + H_{pol}$, where

$$H_{pol}(t) = \frac{\hbar}{2}\int [dp]\lambda(\vec{p},t)[a^\dagger(\vec{p},t)b^\dagger(-\vec{p},t) - b(-\vec{p},t)a(\vec{p},t)] \tag{16}$$

describes the dynamics of vacuum polarization.

Now one can obtain the KE. Let us introduce the distribution functions for the electrons and the holes,

$$
\begin{aligned}
f^e(\vec{p},t) &= \langle\text{in}|a^+(\vec{p},t)a(\vec{p},t)|\text{in}\rangle, \tag{17} \\
f^h(\vec{p},t) &= \langle\text{in}|b^+(-\vec{p},t)b(-\vec{p},t)|\text{in}\rangle. \tag{18}
\end{aligned}
$$

The averaging procedure here is carried out under the in-vacuum state $|\text{in}\rangle$. Differentiation with respect to time and taking into account Equations (14) and (15) results in

$$\dot{f}^e(\vec{p}, t) = \frac{i\lambda}{2}(\vec{p}, t)\left\{f^{(+)}(\vec{p}, t) - f^{(-)}(\vec{p}, t)\right\}, \tag{19}$$

where anomalous averages have been introduced

$$
\begin{aligned}
f^{(+)}(\vec{p}, t) &= \langle\text{in}|a^+(\vec{p}, t)b^+(-\vec{p}, t)|\text{in}\rangle, &(20)\\
f^{(-)}(\vec{p}, t) &= \langle\text{in}|b(-\vec{p}, t)a(\vec{p}, t)|\text{in}\rangle. &(21)
\end{aligned}
$$

The equations of motion for these functions have the form

$$\dot{f}^{(+)}(\vec{p}, t) = \frac{2i}{\hbar}\varepsilon(\vec{p}, t)f^{(+)}(\vec{p}, t) - \frac{i\lambda(\vec{p}, t)}{2}[1 - 2f(\vec{p}, t)], \tag{22}$$

$$\dot{f}^{(-)}(\vec{p}, t) = \frac{-2i}{\hbar}\varepsilon(\vec{p}, t)f^{(-)}(\vec{p}, t) + \frac{i\lambda(\vec{p}, t)}{2}[1 - 2f(\vec{p}, t)]. \tag{23}$$

Here it was assumed that $f^e = f^h = f$ holds as a consequence of the electroneutrality condition.

Let us rewrite Equations (22) and (23) in integral form. Substitution of this result in Equation (19) leads to a KE of non-Markovian type

$$\dot{f}(\vec{p}, t) = \frac{1}{2}\lambda(\vec{p}, t)\int_{t_0}^{t} dt'\lambda(\vec{p}, t')\left[1 - 2f(\vec{p}, t')\right]\cos\theta(t, t'), \tag{24}$$

where

$$\theta(t, t') = \frac{2}{\hbar}\int_{t'}^{t} dt''\varepsilon(\vec{p}, t'') \tag{25}$$

is the dynamical phase.

In the present work the KE (24) and its reformulation in the form of an equivalent system of ordinary differential equations (ODE), shown below in Equation (27), are considered only for zero initial conditions, $f_0 = f(t_0) = 0$. For the first time a KE of such type was obtained in the works [6,7,19] in $D = 3 + 1$ QED for the description of vacuum creation of electron-positron pairs under the action of a time dependent spatially homogeneous linearly polarized electric field. This method is based on the usage of unitary nonequivalent canonical transformations for the transition to the quasiparticle representation [6]. In the considered situation this approach is applicable and leads to the KE (24) that has the same mathematical structure as in the $D = 3 + 1$ QED case [7,8,18]. However, in the massless $D = 2 + 1$ QED case the transition to the quasiparticle representation is possible in the framework of a unitary transformation [15] (see, e.g., Equation (6)).

An advantage of the unitary approach is also the possibility of a generalization of this method [15] to the case of a two-dimensional electric field with the vector potential $A^k(t)(k = 1, 2)$. Let us remark that the transition from the one-dimensional electric field (linear polarization) to two or three field dimensions (arbitrary polarization) in $D = 3 + 1$ QED is connected with the necessity to take into account a larger number of spin degrees of freedom and is accompanied with a significant increase in the number of necessary KE's [20–22].

The main feature of the KE (24) is the absence of an energy gap in the quasienergy (8). Such kind of models were considered long ago [23] (see also [6]) and have been investigated sufficiently well. In the following this feature will be investigated in the situation when the e-h-system in graphene

is exposed to a time dependent electric field. In the presence of the external field the Dirac points $\varepsilon_0(p) = 0$ are transformed to a family of Dirac lines $L_D$ which depend parametrically on time,

$$P_i^D = p_i^D - \frac{e}{c} A_i(t) = 0, \quad i = 1, 2. \tag{26}$$

For the numerical analysis of the KE (24) for different field models it is appropriate to rewrite it in the form of an equivalent system of ODEs [6,8],

$$\dot{f} = \frac{1}{2} \lambda u, \quad \dot{u} = \lambda (1 - 2f) - \frac{2\varepsilon}{\hbar} v, \quad \dot{v} = \frac{2\varepsilon}{\hbar} u, \tag{27}$$

with the corresponding initial conditions $f(t_0) = u(t_0) = v(t_0) = 0$. The auxiliary functions $u(\vec{p}, t)$ and $v(\vec{p}, t)$ describe polarization effects (Section 3) and can be expressed via the anomalous averages (20) and (21)

$$u = \frac{i}{2} \left[ f^{(+)} - f^{(-)} \right], v = \frac{1}{2} \left[ f^{(+)} + f^{(-)} \right]. \tag{28}$$

A concrete physical interpretation of these functions will be given in Section 3.

For the system of Equation (27) one readily obtains the integral of motion

$$(1 - 2f)^2 + u^2 + v^2 = 1, \tag{29}$$

which is compatible with the zero initial conditions.

There is an approximate nonperturbative solution [24] of the KE (24) which is valid for small occupation numbers, $2f \ll 1$ (low density approximation),

$$f_{LD}(\vec{p}, t) \quad = \quad J(\vec{p}, t) = \frac{1}{2} \int_{t_0}^{t} dt' \lambda(\vec{p}, t') \int_{t_0}^{t'} dt'' \lambda(\vec{p}, t'') \cos \theta(t', t''). \tag{30}$$

This integral plays an important role in the formulation of the other nonperturbative approach based on the Markovian approximation (see below).

The polarization function

$$u(\vec{p}, t) = \int_{t_0}^{t} dt' \lambda(\vec{p}, t')[1 - 2f(\vec{p}, t')] \cos \theta(t, t') \tag{31}$$

is transformed in the low density approximation to the quadrature formula

$$u_{LD}(\vec{p}, t) = \int_{t_0}^{t} dt' \lambda(\vec{p}, t') \cos \theta(t, t'). \tag{32}$$

From the low density approximation formula (30) and Equation (10) it follows that the distribution function tends to infinity when approaching the Dirac line, $\vec{p} \to \vec{p}^D(t)$. This indicates also the non applicability of the standard perturbation theory. Thus, close by the lines $L_D$ an essentially nonperturbative analysis of the KE (24) is required. One such nonlinear approximate solution is obtained in the Markovian approximation based on the neglect of the retardation on the r.h.s. of the KE (24), $f(\vec{p}, t') \to f(\vec{p}, t)$. This results in the quadrature formula

$$f_M(\vec{p}, t) = \frac{1}{2} \left\{ 1 - \exp \left[ -2J(\vec{p}, t) \right] \right\}, \tag{33}$$

which has its analog in the case of D=3+1 QED [24]. From here one can see that the distribution function tends to saturation, $f_M(\vec{p}, t) \to 1/2$, at $\vec{p} \to \vec{p}^D(t)$. The polarization function $u(\vec{p}, t)$ can be obtained in this approximation on the basis of the first equation of the system (27) and Equation (33)

$$u_M(\vec{p}, t) = \exp\left[-2J(\vec{p}, t)\right] \int\limits_{t_0}^{t} dt' \lambda(\vec{p}, t') \cos\theta(t, t').\tag{34}$$

where $J(\vec{p}, t)$ is defined by Equation (30).

### 2.2. Order Parameter

By analogy with the standard QED [10], let us introduce the function $\Phi(t) = u(t) + iv(t)$ as an order parameter of the system that describes polarization effects in graphene by means of the anomalous averages (20), (21) and (28) which are characteristic for systems with broken symmetry (e.g. [5,6,16]). We write the corresponding equation of motion

$$\dot{\Phi} - \frac{2i\varepsilon}{\hbar}\Phi = \lambda(1 - 2f),\tag{35}$$

which follows from Equation (27). The formal solution of this equation with the zero initial condition is

$$\Phi(t) = \int\limits_{t_0}^{t} dt' \lambda(t')\left[1 - 2f(t')\right] \exp\left[\frac{2i}{\hbar}\int\limits_{t'}^{t} d\tau\varepsilon(\tau)\right].\tag{36}$$

Let us consider now a finite electric field which is switched off at the point of time $t_{\text{off}}$, i.e., $E(t > t_{\text{off}}) = 0$ and hence according to Equation (10) $\lambda(t > t_{\text{off}}) = 0$. Then, for $t > t_{\text{off}}$ it follows from Equation (36) that the order parameter is different from zero and oscillates with the frequency $2\varepsilon_{\text{out}}/\hbar$, i.e.,

$$\Phi(t > t_{\text{off}}) = \Phi_{\text{out}}(\vec{p}) \exp\left[\frac{2i\varepsilon_{\text{out}}}{\hbar}(t - t_{\text{off}})\right],\tag{37}$$

where the asymptotical value of the quasienergy (8) is equal to

$$\varepsilon_{\text{out}} = \varepsilon(t \to \infty) = v_F \sqrt{\left(\vec{p} - \frac{e}{c}\vec{A}_{\text{out}}\right)^2},\tag{38}$$

$A_{\text{out}}^k = \lim\limits_{t\to\infty} A^k(t)$. In Equation (37) the momentum dependent amplitude

$$\Phi_{\text{out}}(\vec{p}) = \int\limits_{t_0}^{t_{\text{off}}} dt' \lambda(t')\left[1 - 2f(t')\right] \exp\left[\frac{2i}{\hbar}\int\limits_{t'}^{t_{\text{off}}} d\tau\varepsilon(\tau)\right]\tag{39}$$

is defined by the entire prehistory of the system evolution in a given external field.

The presence of such residual oscillations of the order parameter is a prerequisite for the analogous behavior of the polarization current (see Section 5).

Thus,

$$|\Phi(t > t_{\text{off}})|^2 = |\Phi_{\text{out}}(\vec{p})|^2 = \text{const}\tag{40}$$

after switching off the external field, i.e., the long-lived order is formed.

The amplitude $\Phi_{\text{out}}(\vec{p})$ of oscillations of the order parameter in the residual state can be defined from the integral of motion (29) by rewriting it in the form

$$(1 - 2f_{\text{out}})^2 + |\Phi_{\text{out}}|^2 = 1.\tag{41}$$

The order parameter $\Phi(t)$ reflects the role of anomalous averages in the kinetics of the excitation process in graphene, that can be considered as a peculiar field induced phase transition [1,16]. Some other features of this process in graphene will be considered below.

## 3. Observables

It is straightforward to write expressions for the pair number density

$$n(t) \;\; = \;\; N \int [dp] f(\vec{p}, t). \tag{42}$$

The factor $N$ corresponds to number of species (or flavors) of quasiparticles in graphene [3,14,25]: $N = 4$ in the low energy model and $N = 2$ in the tight binding model.

For exact solutions of the ODE system (27) and correct nonperturbative solutions of the type (33) and (34) it follows from the normalization integral (42) that the distribution function is limited everywhere, $f(\vec{p}, t) \leq 1$. Then both polarization functions $u(\vec{p}, t)$ and $v(\vec{p}, t)$ are limited also everywhere under the integral of motion (29). This conclusion relates to the neighborhood of the Dirac lines (26) and to the ultraviolet behavior of these functions as well.

The current density consists of two components, the conduction and polarization current densities,

$$j_k(t) = j_k^{\text{cond}}(t) + j_k^{\text{pol}}(t). \tag{43}$$

These currents are defined by the distribution function $f(\vec{p}, t)$ and the polarization function $u(\vec{p}, t)$, correspondingly [26].

Firstly we consider the currents in the low-energy model. On the basis of the standard definition of the current density [27] ($k = 1, 2$)

$$j_k(t) \;\; = \;\; -e \frac{\delta H(t)}{\delta A_k(t)} \tag{44}$$

one can obtain for the theory with the Hamiltonian (4) taking into account the flavor number

$$j_k(t) \;\; = \;\; 4 e v_F \int d^2 x \Psi^*(\vec{x}, t)\, \sigma_k \Psi(\vec{x}, t). \tag{45}$$

Going over to the quasiparticle representation with the help of the unitary operator (7) we obtain

$$j_k(t) = 4 e v_F \int [dp] \Phi^{\dagger}(\vec{p}, t)\, U^{\dagger}(t) \sigma_k U(t) \Phi(\vec{p}, t). \tag{46}$$

Taking into account the spinor (11) and the definition (43), one can separate the conduction and polarization currents,

$$j_i^{\text{cond}}(t) \;\; = \;\; 8 \int [dp] v_q^i(\vec{p}, t) f(\vec{p}, t), \tag{47}$$

$$j_i^{\text{pol}}(t) \;\; = \;\; 4 \int [dp] \varepsilon(\vec{p}, t) l_i(\vec{p}, t) u(\vec{p}, t),$$

where $v_q^i(\vec{p}, t) = P_i / \varepsilon(\vec{p}, t)$ and the vector $l_i(\vec{p}, t) = \delta \lambda(\vec{p}, t) / \delta E^i(t)$ is defined by the components

$$l_1(\vec{p}, t) = \frac{e v_F^2 P_2}{\varepsilon^2}, \quad l_2(\vec{p}, t) = -\frac{e v_F^2 P_1}{\varepsilon^2}. \tag{48}$$

One can see from the system (27) and its nonperturbative solutions (33) and (34), that the polarization effects dominate in the leading approximation for weak fields, $\alpha = E/E_0 \ll 1$, i.e., $f \sim \alpha^2$, $u \sim \alpha$ and so it follows that

$$|j^{\text{pol}}(t)| \gg |j^{\text{cond}}(t)|. \tag{49}$$

This conclusion is supported also by direct numerical calculations.

Let us note, that the conduction and polarization currents are not collinear in the general case. In the case of the linearly polarized electric field collinearity of the currents (47) rebuilds. In order to ascertain this fact, let us consider the situation when the electric field acts along the axis $x_1$, $\vec{E}(E_1(t), 0)$. Then $P_2 \to p_2$ and the functions $f(\vec{p}, t)$ and $u(\vec{p}, t)$ are even and odd under reflection $p_2 \to -p_2$, respectively, as it can be seen from the structure of the amplitude (10) and Equation (27). This makes the integrals for $j_2^{\text{cond}}(t)$ and $j_2^{\text{pol}}(t)$ in Equation (47) vanish. In order to investigate the theory we calculate the currents (47) in the framework of the perturbation theory in the minimal leading approximation for relatively small external field, see Appendix A.

From Equation (47) it follows that the function $u(\vec{p}, t)$ determines the vacuum polarization current. The physical meaning of the other polarization function $v(\vec{p}, t)$ is revealed if one considers the total energy density of the quasiparticles including the polarization energy. From Equations (12) and (16) one can obtain $E_{\text{tot}} = E_q + E_{\text{pol}}$, where

$$E_q(t) = 8 \int [dp] \varepsilon(\vec{p}, t) f(\vec{p}, t), \tag{50}$$

$$E_{\text{pol}}(t) = 8 \int [dp] \hbar \lambda(\vec{p}, t) v(\vec{p}, t). \tag{51}$$

Taking the time derivative of the quasiparticle energy $E_q(t)$ (50) one obtains

$$\dot{E}_q(t) = \vec{E}(t)[\vec{j}^{\text{cond}}(t) + \vec{j}^{\text{pol}}(t)] = \vec{E}(t)\vec{j}_{\text{tot}}(t), \tag{52}$$

where the currents $\vec{j}^{\text{cond}}(t)$ and $\vec{j}^{\text{pol}}(t)$ are defined by Equation (47).

On the other hand, let us write the Maxwell equation for the internal electric field $\vec{E}_{in}(t)$ generated by the motion of the *eh*-plasma,

$$\dot{\vec{E}}_{\text{in}}(t) = -\vec{j}_{\text{tot}}(t). \tag{53}$$

On this stage we will imply that the total electric field $\vec{E}_{tot}(t)$ is formed by an external field $\vec{E}(t)$ and an internal field $\vec{E}_{in}(t)$, i.e., $\vec{E}_{\text{tot}} = \vec{E}(t) + \vec{E}_{\text{in}}$. Let us substitute now in Equation (52) the external field $\vec{E}(t)$ by the total field $\vec{E}_{\text{tot}}(t)$. Using here Equation (53), we obtain the conservation law of the energy

$$\frac{d}{dt}\left[E_q(t) + \frac{1}{2}E_{\text{in}}^2\right] = \vec{E}\,\vec{j}_{\text{tot}}(t). \tag{54}$$

So, the work of external electric fields (r.h.s. of Equation (54)) is distributed between the energy of e-h excitation and the internal electric field.

## 4. Residual Currents

Here we consider some nonperturbative effects in graphene which are not sufficiently studied in the standard QED or possess some specific features. We restrict ourselves here to the case of a linearly polarized electric field directed along the axis $x_1$.

Let us begin by investigating the residual currents that persist in graphene after the passage of a strong electric field pulse. In the nondissipative model considered here the conduction current discontinues its evolution and remains constant while the polarization current performs damped oscillations. The character of these oscillations and their damping depends on the form of the electric field pulse, as it follows from Equations (37) and (39).

Thus, some oscillating and damped component will be present in the total residual current. In order to calculate it, we will use the formulas for the polarization currents (47) in the low energy model and their analogues in the tight binding model (below in Section 7) with the corresponding polarization function $u^{\text{out}}(t)$ for $t > t_{\text{off}}$,

$$u^{\text{out}}(t) = Re\,\Phi(t > t_{\text{out}}), \tag{55}$$

where the order parameter $\Phi(t > t_{\text{out}})$ in the out-state is defined by Equations (37)–(39).

According to Equation (37) the frequency of the order parameter is defined by the doubled quasienergy (38). However, while these oscillations are smoothed out upon integration over the momentum space in the polarization current (47), their influence remains quite appreciable, see Figure 1.

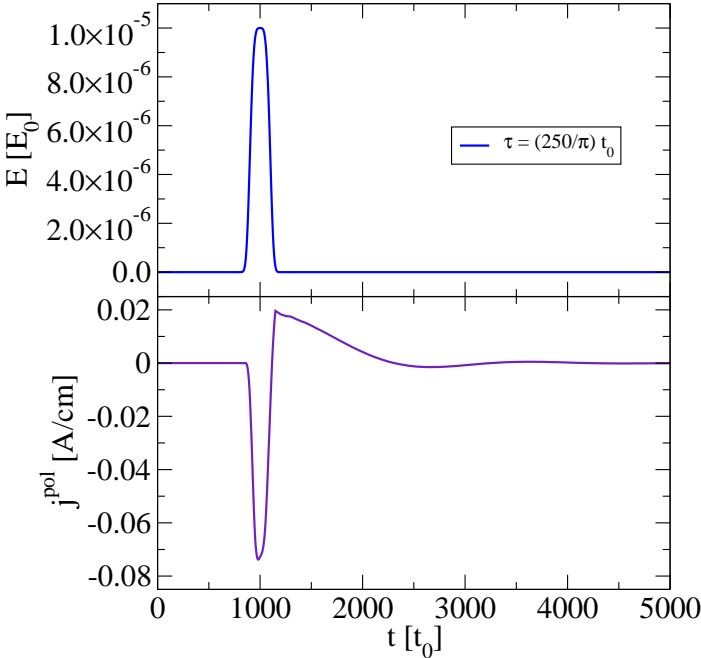

**Figure 1. Upper panel:** Supergaussian electric field (56) where $t_{max} = 1000\, t_0$ ($2.46 \times 10^{-13}$ s) and $E_a = 0.00001\, E_0$ ($1.088 \times 10^3$ V/cm). **Lower panel**: The density of the polarization current.

We select the supergaussian model of the electric field

$$E(t) = -\dot{A}(t) = E_a \exp[-(t - t_{\max})^4 / (2\tau^4)], \tag{56}$$

where $t_{\max}$ determines the position of the maximum amplitude $E_a$ of the field. This choice of the pulse waveform allows to realize abrupt fronts of switching on and off, see the upper panel of Figure 1, and to clearly identify the presence of a alternating polarization current, see the lower panel of Figure 1. This picture demonstrates also dominance of the polarization current.

Another feature of the polarization current becomes apparent in presence of a constant electric field

$$E(t) = E_a = \text{const}, \quad A(t) = -E_a t. \tag{57}$$

Here the oscillations of the polarization function (Figure 2) transform to damped oscillations of the polarization current (Figure 3). This damping is caused by the monotonic growth of the quasienergy (8) with time at $t \geq 0$ and, as a consequence, by the decrease of the oscillation amplitude of the polarization current. This mechanism can be traced visually in the Markovian approximation (33).

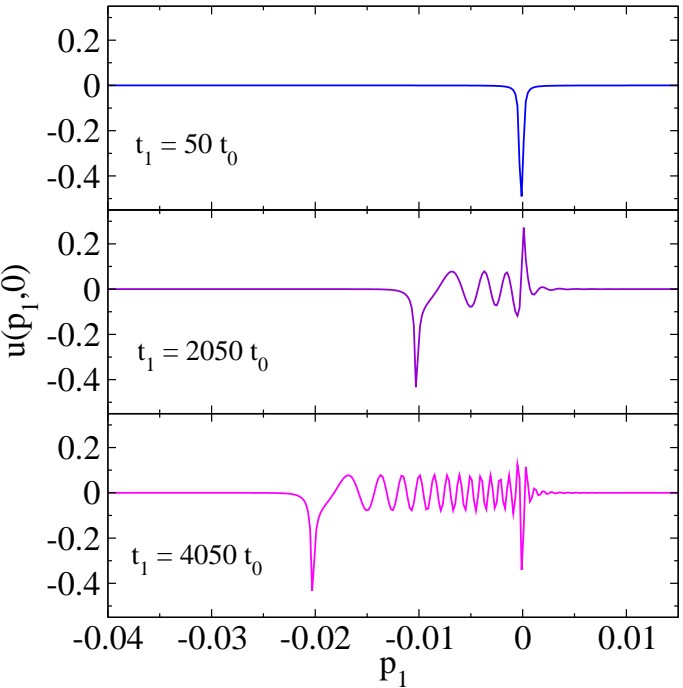

**Figure 2.** The polarization function $u(p_1, p_2 = 0)$ for the times: $t_1 = 50t_0$ ($1.23 \times 10^{-14}$ s, **upper graph**), $t_2 = 2050t_0$ ($5.043 \times 10^{-13}$ s, **middle graph**), $t_3 = 4050t_0$ ($9.963 \times 10^{-13}$ s, **lower graph**) in a constant electric field (57) $E_a = 5 \times 10^{-6} E_0$ ($5.44 \times 10^2$ V/cm).

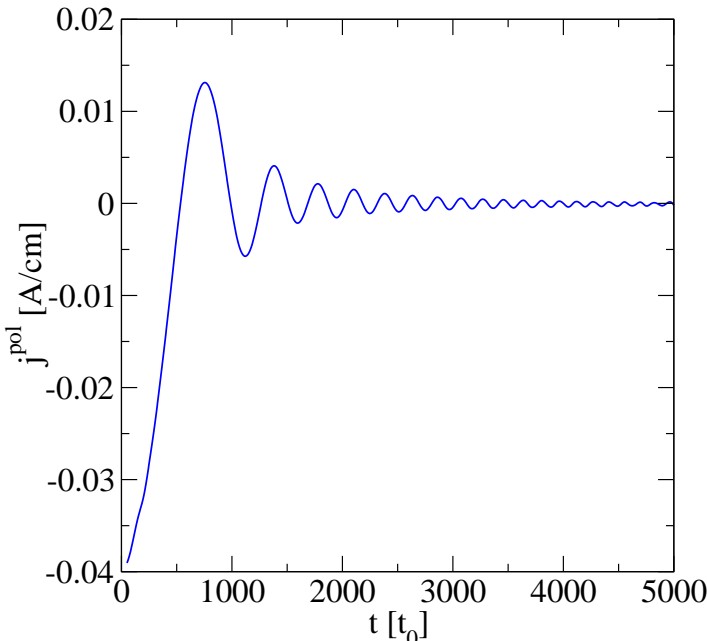

**Figure 3.** The density of the polarization current in a constant field (57) with parameters of Figure 2.

## 5. Numerical Analysis

The numerical analysis will be based on the system of ordinary differential Equation (27) rewritten in terms of the corresponding dimensionless values.

We will investigate the response of the system to four electric field models: the constant electric field (57), the Eckart - Sauter field model

$$E(t) = E_a \cosh^{-2}(t/T), \quad A(t) = -E_a \tanh(t/T), \tag{58}$$

the harmonic function with a constant amplitude

$$E(t) = E_a \sin(\omega t), \quad A(t) = \frac{E_a}{\omega} - \frac{E_a}{\omega} \cos(\omega t), \tag{59}$$

where $\omega$ is the angular frequency, and the "laser field" model [28]

$$\begin{aligned} E(t) &= E_a \cos(\omega t) \exp(-t^2/2\tau^2), \\ A(t) &= -\sqrt{\frac{\pi}{8}} E_a \tau \exp\left(-\sigma^2/2\right) \operatorname{erf}\left(\frac{t}{\sqrt{2}\tau} - i\frac{\sigma}{\sqrt{2}}\right) + c.c., \end{aligned} \tag{60}$$

where $\sigma = \omega\tau$. In all the cases in this section we assume that the electric field is directed along the first coordinate axis.

We start with the most convenient model of the field (58). From Equation (27) follows that the speed of the filling process of the conduction band is determined by the amplitude of the transitions (10). In the denominator of (10) the quasienergy (8) takes zero values on the Dirac line (26). This feature of the amplitude of transitions should be reflected in the behavior and properties of the distribution function. From the form of the evolution of the vector potential (58) it follows that the Dirac line in this case should be represented in the momentum space by a segment with the endpoints determined by $A(t \to -\infty)$ and $A(t \to \infty)$ in accordance with the conditions (26).

In Figure 4 we demonstrate the presence of such characteristic features of the distribution function. On the left panel the Dirac line has the end point coordinates $p_1 \mp 0.1$, $p_2 = 0.0$ while on the right panel the pulse duration is five times larger so that the coordinates of the end points are $p_1 \mp 0.5$, $p_2 = 0.0$. The Dirac line itself cuts in the distribution function a very thin canyon that is not visible in this figure owing to the selected scale of the numerical calculations.

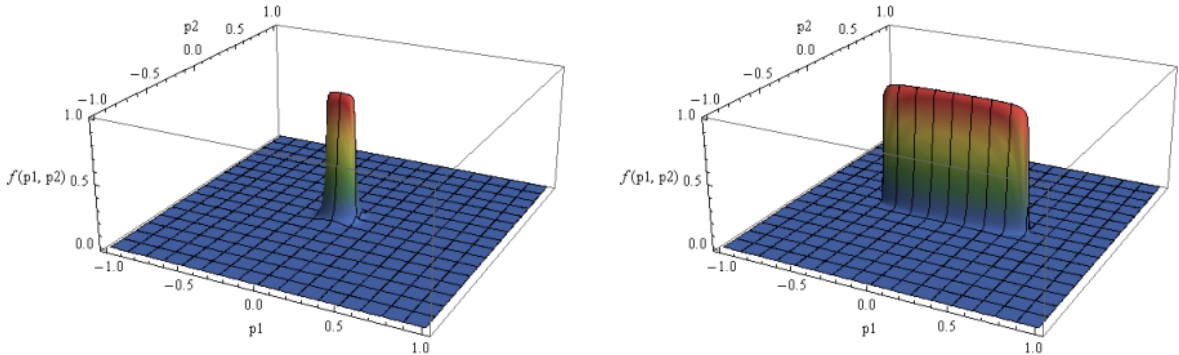

**Figure 4.** The distribution function in the planar momentum space after the action of the Eckart-Sauter pulse (58). **Left panel**: $E_a = 0.01 E_0$ ($1.088 \times 10^8$ V/m) and $T = 10 t_0$ ($2.46 \times 10^{-15}$ s), **Right panel**: $E_a = 0.01 E_0$ ($1.088 \times 10^8$ V/m) and $T = 50 t_0$ ($1.23 \times 10^{-14}$ s).

In the next step, we consider the constant field (57) at $t \geq 0$. The distribution function at the time $t = 10.0 \, t_0$ of the field action is presented on the left panel of Figure 5. Results of the field action with five times longer duration are represented on the right panel of Figure 5.

Another frequently used model is the harmonic electric field (59). The procedure of switching on at $t = 0$ and off at $t_m = 2\pi m/\omega$ can be realized with sufficient accuracy in the numerical calculations. The shape of the distribution function and its change in time ($m = 1, 2, 4$ and $10$) for the field (59) are presented in Figure 6.

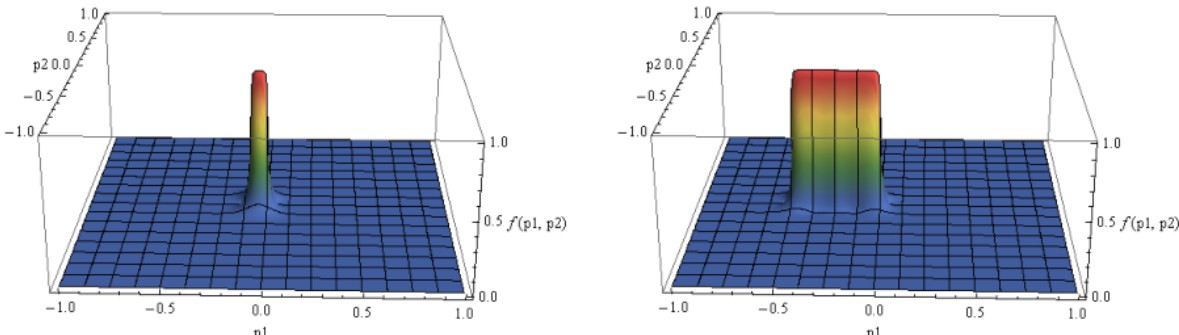

**Figure 5.** The distribution function in the constant electric field (57) with $E_a = 0.01\ E_0$ ($1.088 \times 10^6$ V/cm) at $t = 10\ t_0$ ($2.46 \times 10^{-15}$ s) after switching on the field (**left panel**) and at $t = 50\ t_0$ ($1.23 \times 10^{-14}$ s) after switching on the field (**right panel**).

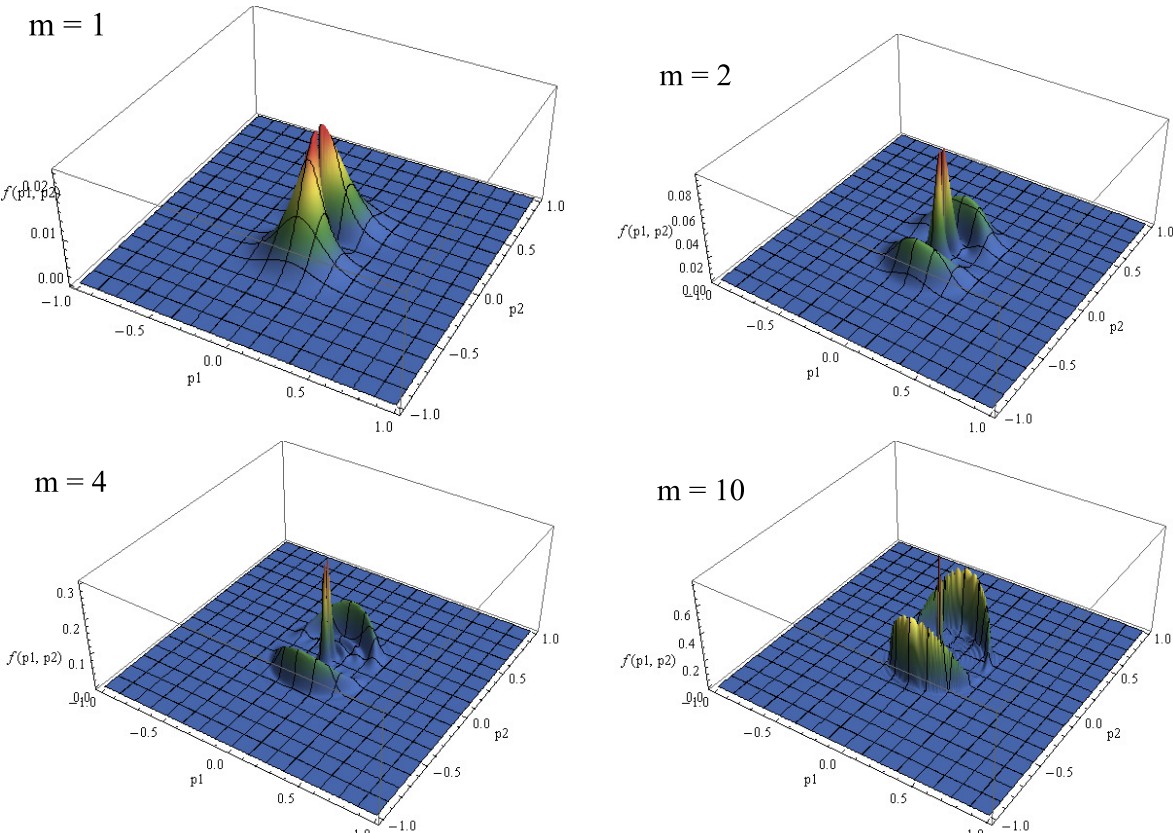

**Figure 6.** The stages of evolution of the distribution function under the action of the field of type (59) with the number of periods $m = 1, 2, 4$ and 10, respectively.

Finally, let us consider the more realistic model (60) of the "laser pulse". In this case the vector potential and the field strength are changed smoothly at any moment of time and do not bear any problems for the numerical calculations. The shape of the distribution function and its dependence on the pulse width determined by parameter $\sigma$ are presented in Figure 7.

The above results correspond to very short time intervals from $T = 10\ t_0$ ($2.46 \times 10^{-15}$ s) to $T = 50\ t_0$ ($1.13 \times 10^{-14}$ s) of the electric field action for the models (57) and (58) and for a very high frequency of oscillations $t_0/T = 0.1 (\approx 400$ THz) for the models (59) and (60). Figure 8 demonstrates the distribution function for the field model (57) and its change for the large time intervals $T = 406{,}500\ t_0$ ($1.0 \times 10^{-10}$ s) and $T = 1{,}219{,}500\ t_0$ ($3.0 \times 10^{-10}$ s) at the field strength $E_a = 9.19 \times 10^{-8}\ E_0 (10$ V/cm). The top row shows images with a linear scale for the color code of the distribution function. This allows to demonstrate that the generated carriers are concentrated in momentum space

in a very narrow area close to zero values of the momentum in the direction perpendicular to the direction of the field. The bottom row shows the same distribution function with a logarithmic scale of the color coding. In this case the complicated structure of the distribution function outside the main area of the carrier generation becomes apparent. The main area, however, is absolutely dominant. Other parameters of the electric field can change this picture. This issue requires further research.

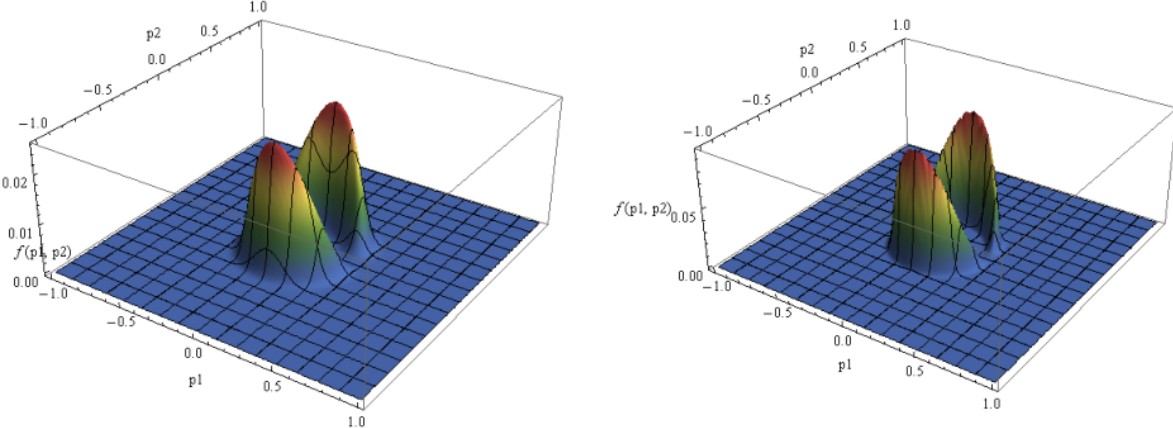

**Figure 7.** The residual distribution function in momentum space after the action of the electric field (60) with $E_a = 0.01\, E_0$ $(1.088 \times 10^6$ V/cm), $\omega = 2\pi 0.1$. **Left:** $\sigma = 5$. **Right:** $\sigma = 10$.

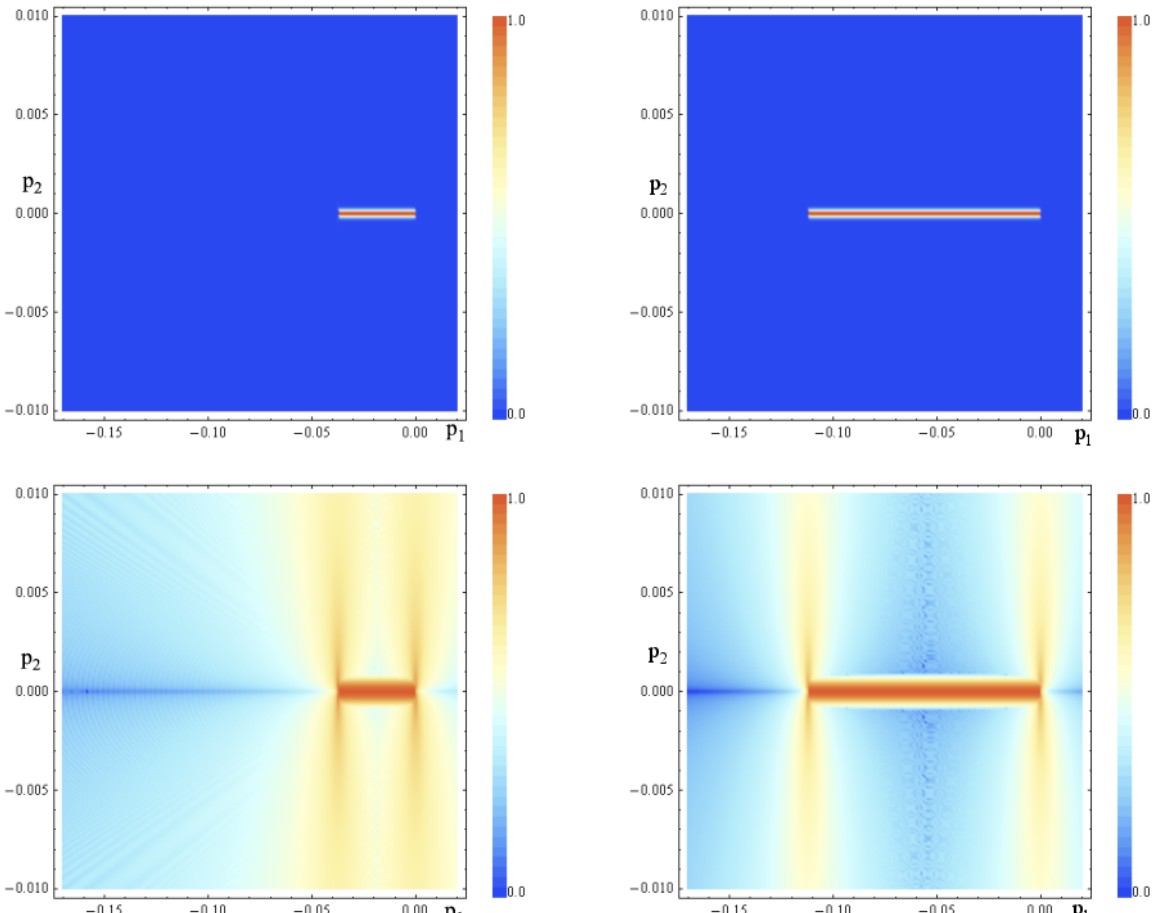

**Figure 8.** The residual distribution function in momentum space after the action of the field of type (57) with $E_a = 10$ V/cm for a duration of $1.0 \times 10^{-10}$ s (**left column**) and $3.0 \times 10^{-10}$ s (**right column**). A logarithmic color scale used for the bottom panel.

The set of parameters used here is quite realizable in the experiment. We note that the behavior of the distribution function in momentum space has not undergone fundamental changes in comparison to the ones in Figure 5. Figure 8 demonstrates agreement with the results of the work [29] where the same parameters have been used for the numerical calculations in the framework of another formalism.

The behavior of a quasiparticle plasma under the action of periodic fields (59) and (60) with increasing pulse duration is not trivial. Figure 9 shows the distribution function for the field model (60) with the parameters $\omega = 2\pi \times 2.46 \times 10^{-4} \, t_0^{-1}$ (corresponding to $2\pi \times 1.0$ THz), $E_a = 9.19 \times 10^{-6} \, E_0$ (corresponding to 1000 V/cm) for $\sigma = 3, 10, 25$, and 50.

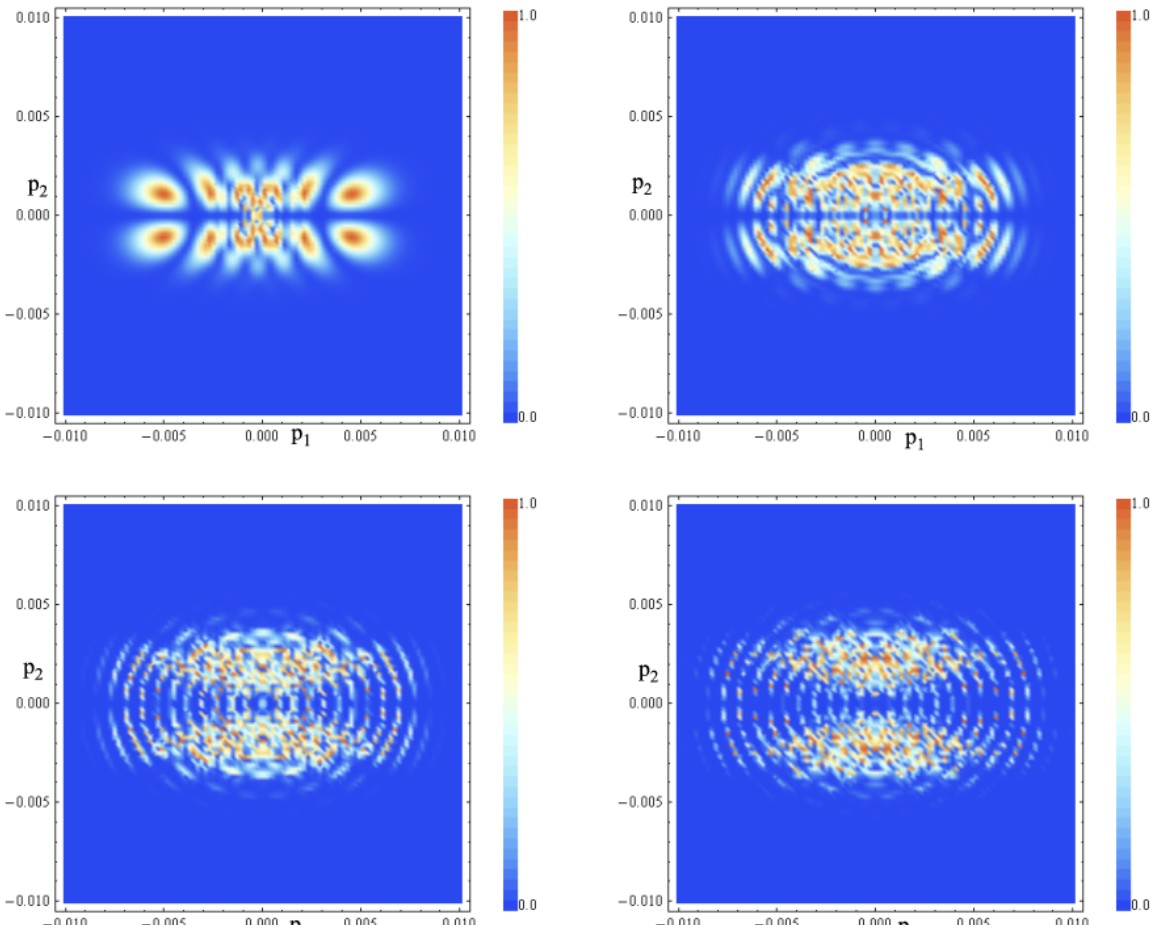

**Figure 9.** The residual distribution function under the action of the field of type (60) with the frequency 1.0 THz, amplitude $E_a = 1000$ V/cm and $\sigma = 3, 10, 25, 50$.

Let us now discuss the analysis of some observable values. We have studied the behavior of the density of carriers (42) in dependence on the amplitude of the electric field for the three models (57), (58) and (60). A summary of the results is presented below.

It should be noted that these results are determined solely by the filling of the conduction band due to the quasiparticle excitation by an external electric field. The presence of thermally excited carriers has not been taken into account as well as the relaxation processes since the considered times are much shorter than the relaxation time. Figure 10 demonstrates that the number of quasiparticles created during the action of the field increases quadratically with increasing electric field strength. The quadratic dependence can be traced quite rigorously in relatively weak fields $E_a \leq 0.001 \, E_0 \, (\lesssim 10^5$ V/cm). The increment of the pair number density is slowing down somewhat with further increase of the electric field. This can be explained by a saturation effect.

The presented values correspond to the pulse of the constant field (57) with duration $20t_0$, the Eckart-Sauter pulse (58) with duration parameter $T = 10\ t_0$ and a "laser" pulse (60) with a period of the carrier frequency equal to $2\pi/\omega = 10\ t_0$. Such a proximity of the characteristics of the compared field pulses provides very similar values for the surface density of the charge carriers. However, it should be noted that above it has been demonstrated that there is a strong difference between the quasiparticle spectrum in the field of type (60) and the quasiparticle spectrum produced by fields of the type (57) or (58). Nevertheless, the density of carriers and their dependence on the amplitude of the electric field are very similar, see Figure 10.

The dotted line in Figure 10 indicates the approximate level of the thermal carrier density at room temperature. For short pulses their contribution to the total number of carriers will be noticeable only at high electric fields. On the other hand, the spectrum of thermal quasiparticles and quasiparticles generated by the field pulse are different. These differences appear at any electric field strength.

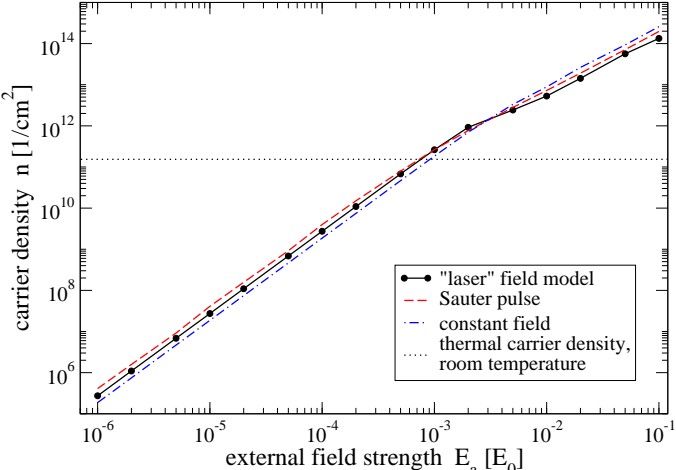

**Figure 10.** The dependence of the carrier density for the electric field models (57), (58) and (60).

Now we come back to the constant field and look at the dynamics of the process of creation of the quasiparticles in the period of the field action. We consider a weak field strength of about 1 V/cm and a field action time of $5 \times 10^5 t_0$ (corresponding to $1.23 \times 10^{-10}$ s). The left panel of Figure 11 shows the evolution of the distribution function along the direction of the electric field (for $p_2 \simeq 0$). The sections of the distribution function along the $p_1$ axis are presented for six time points from $t_1 = 25{,}000\ t_0$ to $t_6 = 500{,}000\ t_0$. This figure shows in more detail the dynamics that we have already seen in Figures 5 and 8. The complete picture is presented in the right panel of Figure 11. It shows the evolution of a slice of the distribution function for the value of $p_1 = -0.002002\ p_0$ in which the distribution function at the initial period ($t_1 = 2.5 \times 10^4 t_0$, $t_2 = 1 \times 10^5 t_0$) is not large. At the time $t_3 = 2 \times 10^5 t_0$ there is a transition in the state of saturation and then the picture becomes almost stationary.

Figure 12 shows the evolution of three observables under the action of a constant electric field with the same parameters as in Figure 11. The left panel shows the time dependence of the density of the charge carriers (42). The dashed line shows the linear extrapolation of the initial values. The middle panel shows the time dependence of the density of the conduction current in the direction of the field. The dashed line also shows a linear extrapolation. It can be concluded that the creation of charge carriers in a weak constant field proceeds at a constant rate. The energy of the carriers is proportional to their momentum. The number of carriers and the average value of their momentum increase under the influence of the field. As a result, the energy density of the carriers in graphene increases quadratically. This is shown in the right panel of Figure 11 (the dashed line shows the quadratic extrapolation of the initial values).

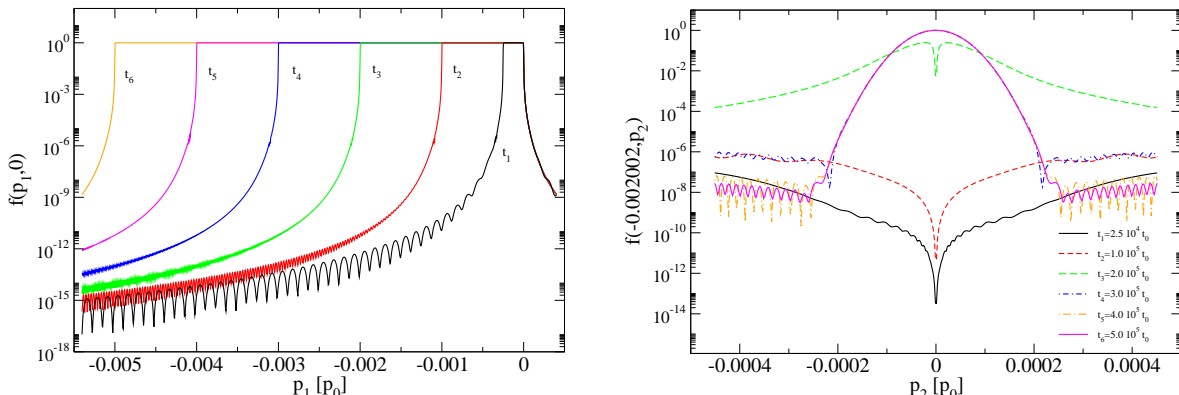

**Figure 11.** The distribution function in momentum space for the range of time: $t_1 = 2.5 \times 10^4 \, t_0$ ($6.15 \times 10^{-12}$ s), $t_2 = 1 \times 10^5 \, t_0$ ($2.46 \times 10^{-11}$ s), $t_3 = 2 \times 10^5 \, t_0$ ($4.92 \times 10^{-11}$ s), $t_4 = 3 \times 10^5 \, t_0$ ($7.38 \times 10^{-11}$ s), $t_5 = 4 \times 10^5 \, t_0$ ($9.84 \times 10^{-11}$ s), $t_6 = 5 \times 10^5 \, t_0$ ($1.23 \times 10^{-10}$ s). The electric field strength $E_a = 1 \times 10^{-8} \, E_0$ (1.088 V/cm). The dependence of the distribution function on $p_1$ for $p_2 = 0$ is shown on the left panel while the dependence on $p_2$ for $p_1 = -0.002002 \, p_0$ is shown on the right panel.

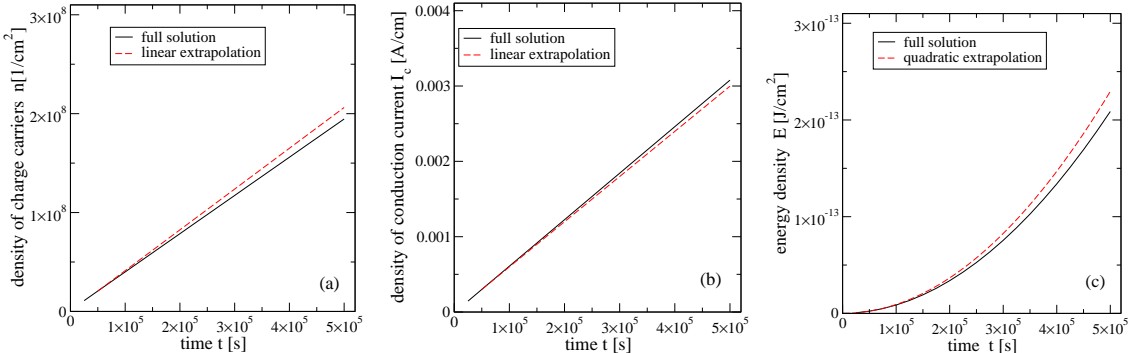

**Figure 12.** From left to right: density of charge carriers (42), density of conduction current and energy density the carriers for range of the time 25,000 $t_0 -$ 500,000 $t_0$ ($6.15 \times 10^{-12}s - 1.23 \times 10^{-10}s$). The constant electric field strength $E_a = 1 \times 10^{-8} \, E_0$ (1.088 V/cm).

## 6. Graphene as Active Medium

The nonlinear properties of graphene allow to activate it by some basic electric field for driving by the current of another weak signal. Below we will consider an example when a rather strong basic field is aligned with the $x_1$—axis while the probe field aligned with the $x_2$—axis allows the application of perturbation theory.

The corresponding perturbation theory can be constructed both on the system of Equation (27) and in the framework of the Markovian approximation (33). The latter variant allows to proceed with analytical calculations.

In the framework of such an approximation we can limit ourselves in the weak field approximation to the Markovian solution (33) and (34). The distribution function in this approximation is $f_M \approx f^{(0)} + f^{(1)}$,

$$f^{(0)} = \frac{1}{2}(1 - e^{-J^{(0)}}), \quad f^{(1)} = J^{(1)} e^{-J^{(0)}}, \tag{61}$$

where labels (0) and (1) correspond to the "basic" field $A_1(t)$ (nonperturbative solutions) and to the perturbing field $A_2(t)$, respectively. The function $J(\vec{p}, t)$ is defined by Equation (33), so that

$$J^{(1)}(\vec{p}, t) = 8 \int\limits_{t_0}^{t} dt' \lambda^{(1)}(\vec{p}, t') \int\limits_{t_0}^{t'} dt'' \lambda^{(0)}(\vec{p}, t'') \cos\theta^{(0)}(t', t''), \tag{62}$$

where according to Equation (10)

$$\lambda^{(0)}(\vec{p},t) = \frac{ev_F^2 E_1(t)p_2}{2[\varepsilon^{(0)}(\vec{p},t)]^2}, \quad \lambda^{(1)}(\vec{p},t) = \frac{ev_F^2 E_2(t)P_1}{2[\varepsilon^{(0)}(\vec{p},t)]^2}. \tag{63}$$

In Equation (62) we have neglected in the phase (25) the frequency shift under the influence of the weak field $A^2(t)$, i.e., $\theta \to \theta^{(0)}$. An analogous decomposition of the polarization function (34) leads to the result: $u_M \simeq u^{(0)} + u^{(1)}$, where

$$u^{(1)}(\vec{p},t) = J^{(0)}(\vec{p},t)\left\{ \int_{t_0}^{t} dt'\lambda^{(1)}(\vec{p},t')\cos\theta^{(0)}(t,t') - J^{(1)}(\vec{p},t)\int_{t_0}^{t} dt'\lambda^{(0)}(\vec{p},t')\cos\theta^{(0)}(t,t') \right\}. \tag{64}$$

Substitution of Equations (61)–(64) into Equation (47) results in the perturbed current calculated in the first order of perturbation theory under the weak field,

$$j_1^{(1)}(t) = 0, \quad j_2^{(1)}(t) = \int_{t_0}^{t} dt'\sigma(t,t')E_2(t'), \tag{65}$$

where $\sigma(t,t')$ is the linear induced conductivity of graphene controlled by the external field $A_2(t)$,

$$\sigma(t,t') = -8ev_F^2 \int \frac{[dp]}{\varepsilon^{(0)}(\vec{p},t)}\left\{ \frac{\delta f^{(1)}(\vec{p},t)}{\delta E_2(t')} + f^{(2)}(\vec{p},t)\frac{ev_F^2 P_1^2(t)}{[\varepsilon^{(0)}(\vec{p},t)]^2} \right.$$
$$\left. + P_1(t)\frac{\delta u^{(1)}(\vec{p},t)}{\delta E_2(t')} - u^{(0)}(\vec{p},t)\frac{ev_F^2 p_2}{[\varepsilon^{(0)}(\vec{p},t)]^2} \right\}. \tag{66}$$

Here the first and second groups of terms correspond to the contributions of the conduction and polarization currents.

The dependence of the conductivity (66) on the magnitude and spectral decomposition of the basic field will be considered separately.

## 7. Tight Binding Model

It is not difficult to obtain now the analogous KE in the $D = 2 + 1$ tight binding model of the nearest neighbor interaction [3,11,12]. The Hamiltonian function in the momentum representation in this case is

$$H_{\vec{p}}(t) = \begin{pmatrix} 0 & h_{\vec{p}}(t) \\ h_{\vec{p}}^*(t) & 0 \end{pmatrix} = h_{\vec{p}}'(t)\sigma_1 - h_{\vec{p}}''(t)\sigma_2, \tag{67}$$

where

$$h_{\vec{p}}(t) = h_{\vec{p}}'(t) + ih_{\vec{p}}''(t) = -\gamma\sum_{\alpha}\exp\left(\frac{i}{\hbar}\vec{P}\vec{\delta}_\alpha\right), \tag{68}$$

with $\gamma \approx 2.7$ eV being the hopping energy, and

$$\vec{\delta}_1 = \frac{a}{3}(0,\sqrt{3}), \quad \vec{\delta}_2 = \frac{a}{3}(\pm 3/2, -\sqrt{3}/2) \tag{69}$$

are the locations of the nearest neighbors, $a \approx 3$.

An external electric field is introduced here according to the rule $\vec{p} \to \vec{P} = \vec{p} - e/c\vec{A}(t)$. Such a method was used in the work [3] in the case of a constant electric field $A_2(t) = -eEt$ and resulted immediately in a nonlinear interaction with external field. Such a theory belongs to the class of theories with the highest derivatives.

The Hamiltonian functions (4) and (67) have the same pseudospin structure. Therefore one can follow the way of derivation of KE (24) in the theory with the Hamiltonian function (67). The quasienergy (8) and the amplitude (10) are changed only by the following formal substitutions

$$v_F P_1 \to h'_{\vec{p}}(t), \ v_F P_2 \to -h''_{\vec{p}}(t). \tag{70}$$

This results in

$$\varepsilon(\vec{p}, t) = \sqrt{h^*_{\vec{p}}(t) h_{\vec{p}}(t)} = |h_{\vec{p}}(t)|, \tag{71}$$

$$\lambda(\vec{p}, t) = \frac{1}{|h_{\vec{p}}(t)|^2} \left\{ h''_{\vec{p}}(t) \dot{h}'_{\vec{p}}(t) - \dot{h}'_{\vec{p}}(t) h''_{\vec{p}}(t) \right\}. \tag{72}$$

An analogous KE can be obtained also for the case of the multilayer graphene model [30].

The conduction and polarization currents have the following form (the flavor number in the given model is equal to 2 [3])

$$j_k^{\text{cond}}(t) = -4e\gamma \int [dp] f(\vec{p}, t) [F_k^{(1)}(\vec{p}, t) \cos \chi + F_k^{(2)}(\vec{p}, t) \sin \chi], \tag{73}$$

$$j_k^{\text{pol}}(t) = -4e\gamma \int [dp] f(\vec{p}, t) [-F_k^{(1)}(\vec{p}, t) \sin \chi + F_k^{(2)}(\vec{p}, t) \cos \chi], \tag{74}$$

with the vector formfactors

$$F_k^{(1)}(\vec{p}, t) = \sum_\alpha \delta_\alpha^{(k)} \sin \left( \frac{1}{\hbar} \vec{P} \vec{\delta}_\alpha \right), \tag{75}$$

$$F_k^{(2)}(\vec{p}, t) = \sum_\alpha \delta_\alpha^{(k)} \cos \left( \frac{1}{\hbar} \vec{P} \vec{\delta}_\alpha \right) \tag{76}$$

and $\chi$ being the angle of the unitary rotation in the matrix of the type (7),

$$\chi = -h''_{\vec{p}}(t) / h'_{\vec{p}}(t). \tag{77}$$

Let us rewrite Equations (73) and (74) for the currents to obtain

$$j_k^{\text{cond}}(t) = -4e\gamma \int [dp] f(\vec{p}, t) \sum_\alpha \delta_\alpha^{(k)} \sin \left( \chi + \frac{1}{\hbar} \vec{P} \vec{\delta}_\alpha \right), \tag{78}$$

$$j_k^{\text{pol}}(t) = -4e\gamma \int [dp] u(\vec{p}, t) \sum_\alpha \delta_\alpha^{(k)} \cos \left( \chi + \frac{1}{\hbar} \vec{P} \vec{\delta}_\alpha \right) \tag{79}$$

as the final result of this section. The numerical evaluation of the currents for particular external field models is delegated to future work.

## 8. Conclusions

We have obtained on a nonperturbative basis the KE for describing electron-hole excitations in graphene under the action of a spatially homogeneous time dependent electric field. To this end the analogy with the well-developed case of kinetic theory of vacuum $e^+ e^-$ plasma generation in strong fields [8,10] in $D = 3 + 1$ QED has been used. As a rule, we used the simplest low energy model. However, the used method admits a straightforward generalization to other realistic models of the carrier dynamics as, e.g., the tight binding model of nearest neighbour interaction. The derivation of the KE is based on the transition to the quasiparticle representation [6]. As shown in Section 2, in the $D = 2 + 1$ QED model of graphene such a derivation can be given in an explicit form with the help of a unitary transformation first introduced in the work [15] for the linearly polarized electric field. It is important that the final KE is valid in the general case of an arbitrarily polarized electric field. Some features of the obtained KE are discussed in that Section. In particular, the non applicability of

the standard perturbation theory in vicinity of the Dirac lines has been demonstrated. However, the corresponding approximate analytical and numerical nonperturbative solutions (e.g., the Markovian approximation) of the KE provide a correct description in physical terms in the entire momentum space. In Section 2.2 we consider also some general properties of the evolution of the excited electron-hole plasma that allow to interpret this phenomenon as a specific field induced phase transition [10,16]. An important characteristics of this process is an order parameter that continues to oscillate in the out-state after the external field pulse ceases. The connection of the observables with both, the distribution function and the polarization functions has been discussed in Section 3. The damped oscillations of the residual polarization current on the background of a conduction current were considered in Section 4. The character of these oscillations is related to features of the external field pulse. The damped oscillations of the polarization current in a constant electric field have demonstrated a similar nature. Apparently, these effects are accessible to experimental observation. It can be assumed that similar phenomena occur in $D = 3 + 1$ QED. Here too it was shown that the polarization current dominates over the conduction current. We have performed a systematic numerical investigation based on the KE for the distribution function of quasiparticle excitations and the corresponding observable values for various models of the external electric field.

We have discussed the possibility of using graphene as an active medium excited by the basic electric field to be probed by another signal current ("pump-and-probe").

Finally, we have derived an analogous KE for the tight binding model that is substantially nonperturbative. In the framework of this model we have obtained and discussed the conduction and polarization currents.

A verification of the developed theory was obtained in the work [31], where good agreement with experiment has been shown for the case of a constant electric field [32].

Let us note that both models considered here led KE's of identical form. Moreover, this form invariance is conserved also in the standard QED in the case of a linearly polarized electric field when the spin degrees of freedom are frozen.

**Author Contributions:** Conceptualization, S.A.S. and A.D.P.; Validation, all authors; Data Curation, A.D.P. and N.T.G.; Writing—Original Draft Preparation, S.A.S. and A.D.P.; Writing—Review & Editing, D.B.B. and N.T.G.; Visualization, A.D.P. and N.T.G.; Funding Acquisition, S.A.S. and D.B.B.

**Funding:** This research was supported in part by the "RUDN University Program 5-100", by RFBR according to the research project No 17-02-00375 A, and by the Polish NCN under grant number UMO-2014/15/B/ST2/03752.

**Acknowledgments:** The authors thank V.V. Dmitriev, B. Dora, D.M. Gitman and R. Moessner for useful discussions. D.B. is grateful to Hayk Sarkisyan for inspiring discussions on low-dimensional quantum systems and for the hospitality extended to him at the Russian-Armenian University.

**Conflicts of Interest:** The authors declare no conflict of interest.

## Appendix A. Perturbation Theory

In order to demonstrate the effectiveness of the introduced approach, we will reproduce some well known results in the framework of perturbation theory for relatively small external fields, $E < E_0$.

We begin with the analysis of currents in the low density approximation (30) and (32) which corresponds to the one-photon excitation mechanism [33]. In the minimal leading approximation $\varepsilon(\vec{p}, t) \to \varepsilon_0(\vec{p}) = v_F|\vec{p}|$ we have

$$f_{LD}^{(2)}(\vec{p}, t) = \frac{1}{2} \int^t dt' \lambda^{(1)}(\vec{p}, t') , u_{LD}^{(1)}(\vec{p}, t') \tag{A1}$$

$$u_{LD}^{(1)}(\vec{p}, t) = \int^t dt'' \lambda^{(1)}(\vec{p}, t'') \cos[\eta p(t - t'')], \tag{A2}$$

with

$$\lambda^{(1)}(\vec{p}, t) = \lambda_0(\vec{p}) \left[ E_1(t) P_2 - E_2(t) P_1 \right] , \tag{A3}$$

where $\lambda_0(\vec{p}) = ev_F^2/2\varepsilon_0^2(p) = e/2p_2$ and $\eta = 2v_F/\hbar$. The upper indices at the functions $f^{(2)}$ and $u^{(1)}$ indicate the order of perturbation theory.

The relations (A1) and (A2) indicate the dominant role of polarization effects in the considered approximation, $|j^{(1)\text{pol}}(t)| \gg |j^{(2)\text{cond}}(t)|$.

Let us consider the case of linear polarization $\vec{E}(t) = (E(t), 0)$ with arbitrary time dependence of the electric field.

The polarization current in lowest order perturbation theory according to Equation (47) is

$$
\begin{aligned}
j_1^{(1)\text{pol}}(t) &= 8ev_F \int [dp]u \sin\varkappa, \\
j_2^{(1)\text{pol}}(t) &= -8ev_F \int [dp]u \cos\varkappa,
\end{aligned}
\tag{A4}
$$

where $\varkappa$ is defined in explanation to Equation (7),

$$
\varkappa = \arctan(P_2/P_1) \approx \arctan(p_2/p_1),
\tag{A5}
$$

where the last step corresponds to the leading approximation. From Equation (A5) follows

$$
\sin\varkappa \approx p_2/p = \sin\Phi, \quad \cos\varkappa \approx p_1/p = \cos\Phi,
\tag{A6}
$$

where $\Phi$ is the polar angle in the polar representation of the momentum space. Integration over the momentum $p$ in the neighborhood of the Dirac points is limited by the cutoff parameter $\Lambda$. It is implied that it can be defined by the limits of the validity range of the linear dispersion law $\varepsilon_0(p) = v_F p$. However, the results obtained below are universal and do not depend on the choice of $\Lambda$.

Taking these remarks into account, one can thus write the polarization current after integration over the angle (here $t_0 \to -\infty$),

$$
\begin{aligned}
j_1^{(1)\text{pol}}(t) &= \frac{e^2 v_F}{\pi \hbar^2} \int_{-\infty}^{t} dt' E(t') \int_0^\Lambda dp \cos[\eta p(t-t')], \\
j_2^{(1)\text{pol}}(t) &= 0 .
\end{aligned}
\tag{A7}
$$

Let us now perform a Fourier transformation of the function $E(t)$ and after that integrate over the momentum $p$,

$$
\begin{aligned}
j_1^{(1)\text{pol}}(t) &= \frac{e^2}{\pi\hbar} \int d\omega E(\omega) \int_{-\infty}^{t} dt' \frac{\sin[\Lambda\eta(t-t')]}{t-t'} e^{i\omega t'} \\
&= \frac{e^2}{\pi\hbar} \int d\omega E(\omega) e^{i\omega t} \int_{-\infty}^{0} \frac{dx}{x} \sin(\gamma x) \cos x.
\end{aligned}
\tag{A8}
$$

The last integral does not depend on the parameter $\gamma = 2v_F\Lambda/\hbar\omega$,

$$
\int_{-\infty}^{0} \frac{dx}{x} \sin\gamma x \cos x = \int_{-\infty}^{0} \frac{dx}{x} = \frac{\pi}{2} \, ,
\tag{A9}
$$

so that

$$
j_1^{(1)\text{pol}}(t) = \frac{e^2}{4\hbar} E(t).
\tag{A10}
$$

This result does not depend on the choice of the field model.

In order to calculate the conduction current, it is necessary to find the distribution function. To this end we use perturbation theory as a first step and consider the case of a constant electric field (57) switched on at the time $t_0 = 0$. In the leading approximation from Equation (30) follows the known result [15,34]

$$f_{LD}^{(2)}(\vec{p}, t) = \frac{e^2 \hbar^2 E^2 p_2^2}{4 v_F^2 p^6} \sin^2 \Omega t \,, \tag{A11}$$

where $\Omega = v_F p / \hbar$ is the frequency of the vacuum oscillations.

The anisotropic distribution (A11) and the corresponding nonperturbative Markovian distribution (33) have the center symmetry relative to the Dirac point $p_i \to -p_i$ whereby the conductivity current (see Equation (47)) vanishes.

In order to break this symmetry, it is necessary to go beyond the leading approximation. However, the next correction leads to secular terms that indicate a problem with perturbation theory. For further details on the transport properties of graphene see, e.g., Refs. [35–39].

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
