# Peer review of "Nonperturbative Kinetic Description of Electron-Hole Excitations in Graphene in a Time Dependent Electric Field of Arbitrary Polarization"

_2571-712X, doi:10.3390/particles2020015_

Round 1

Reviewer 1 Report

Referee's report on particles-418432

"Nonperturbative kinetic description ..."

by S. A. Smolyansky et al.

The authors transfer their expertise from the dynamical Schwinger effect

in QED to the treatment of electron hole excitations in a graphene layer

exposed to an electric field. By using extensively the parallels of both

systems a formalism is elaborated and analyzed which addresses, in

particular, the nonperturbative features. As such, the paper is a suitable

contribution to the announced special issue of MDPI-particles. I fully

support the publication.

In preparing the final version the authors may consider the following

issues.

1) At a few places, the formulation is in bumpy English (this can be

easily improved). I also see a few typos (use a spell checker).

2) Be careful with acronyms (e.g. in the Introduction "e-h pair creation"

is mentioned w/o introducing the notion of "e-h pair").

3) The authors introduce the quantity Phi (cf. (25)) as THE order

parameter of the system. Provide proper references to demonstrate

that this the only one or among (many?) possible other ones.

4) Time derivatives are depicted by dots over the respective symbols.

In (52, 53), however, I see bullets (misprint or a new special

operation?).

5) In a few figure captions the authors claim to display "the evolution

of ...". Instead, I see a series of snapshots, e.g. figs. 6 and 9.

Sometimes, it is unclear which instant of time is selected, e.g. figs. 7

and 9.

6) What is the meaning of the bottom vs. top panels in fig. 8?

7) "n" in fig. 6 is a number, while in (42) and fig. 10 it denotes the

density.

8) The Introduction announces to derive in § 7 some currents (what

is done, in fact), but their relation to the paper's content is unclear.

It seems that this section can be dropped w/o any consequence for

the otherwise nice paper.

Author Response

The authors are grateful to the referee for the careful study of our manuscript and for the positive recommendation.We thank for the insightful comments and critical remarks that helped us to improve the quality of the manuscript. Below we give point-by-point replies to the referees comments and explain the corresponding changes made in the manuscript (in red color).

In preparing the final version the authors may consider the following

issues.

1) At a few places, the formulation is in bumpy English (this can be

easily improved). I also see a few typos (use a spell checker).

We have performed a throrough revision of the English language of the manuscript,

replaced the bumpy formulations and elimiated typos.

2) Be careful with acronyms (e.g. in the Introduction "e-h pair creation"

is mentioned w/o introducing the notion of "e-h pair").

We introduced the abbreviation “e-h” for “electron-hole” at its first occurrence in 

the Introduction.

3) The authors introduce the quantity Phi (cf. (25)) as THE order

parameter of the system. Provide proper references to demonstrate

that this the only one or among (many?) possible other ones.

We admit that other definitions of an order parameter are possible and thus we changed the phrase in the first sentence of subsection 2.2. accordingly. We provided references for the choice suggested in this manuscript.

4) Time derivatives are depicted by dots over the respective symbols.

In (52, 53), however, I see bullets (misprint or a new special operation?).

Bullets were used for better visibility only. We changed back to dots.

5) In a few figure captions the authors claim to display "the evolution

of ...". Instead, I see a series of snapshots, e.g. figs. 6 and 9.

Sometimes, it is unclear which instant of time is selected, e.g. figs. 7

and 9.

Indeed, fig. 6 shows several successive snapshots of the distribution function for the field model (59). These are the moments of time when the field takes a zero value (one of the consecutive field shutdowns).

This is explained in some detail in the text.

We changedthe caption to this figureto: “The stages of evolution of the distribution function…”.

In Figure 9, of course, it is not the evolution of the distribution function, or even not the sequence of snapshots of the distribution function for the field model (60). 

These are the distribution functions for a finite point in time after the field is completely discontinued due to Gaussian clipping (as in fig. 7). Fig. 7 simply demonstrates two examples of residual distribution of carriers for this field model. 

We changedthe caption tofig.to: “The residual distribution function in momentum space…”.

The original meaning of Figure 9 was to show that an increase in the number of periods of the field under a Gaussian envelope leads to a complication of the form of the residual distribution function of the carriers.

In this sense, the situation is somewhat similar to the change in the distribution function with an increase in the number of field periods in model (59).

And this was the reason for usingthe term evolutionin a not very direct sense.

We changed the caption of fig. 9 to: “The residual distribution function under the action…”.

The word residualhasbeenadded to the caption for Figure 8.

6) What is the meaning of the bottom vs. top panels in fig. 8?

On the bottom panel of Figure 8 logarithmic color scale is used.

This allows to detect some details of the behavior of the distribution function which are not visible in the top panel.

First of all, it becomes obvious that small perturbations of the distribution function are not localized. In this case, it does not make a significant contribution to the integral characteristics, but this is useful to keep in mind.

We added to the figure caption the word “residual” and the sentence: “A logarithmic color scale is used for the bottom panel”.

7) "n" in fig. 6 is a number, while in (42) and fig. 10 it denotes the

density.

We changed n → m in Fig. 6 and in the corresponding part of the text.

8) The Introduction announces to derive in § 7 some currents (what is done, in fact), but their relation to the paper's content is unclear. It seems that this section can be dropped w/o any consequence for the otherwise nice paper.

We give in section 7 the application of the formalism for the tight-binding case and derive the expressions for conduction and polarization currents which are potentially accessible to observation. At the end of this section we added the remark that the numerical evaluation of this part for specifica field models is delegated to future work.

Reviewer 2 Report

@page { margin: 0.79in } p { margin-bottom: 0.1in; line-height: 120% } a:link { so-language: zxx }

In the manuscript a nonperturbative framework based on a Dirac-type equation is used to describe electron-hole excitaions in graphene under strong time-dependent electric fields. Various models and approximations of the dynamics are considered and the results indicate that the efficiency of the e-h-excitations increases when the field polarization is changed from linear to elliptic. Overall, I believe that the results of the manuscript are significant enough to warrant publication. However, I have a few questions that should be addressed and clarified by the authors:

The initial condition (after Eq. (25)) of the kinetic equation chosen in this paper is an assumption. What is the physical justification of this (if there is any?) and what would change if different initial conditions are chosen? For instance the amplitude (39) depends explicitly on the initial condition.

Various models for the electric field have been chosen and strong model dependencies are observed for the distribution function, however the carrier density is surprisingly model independent. Do the authors have an explanation for this? If so, it should be mentioned in the manuscript.

I am a bit confused by the statement on page 18 “The corresponding perturbation theory can be constructed both on the system of equations (27) and in the framework of the Markovian approximation (33).” I was under the impression that the Markovian approximation is essentially nonperturbative. This needs to be clarified.

I was wondering whether the authors have considered the use of dynamical equations other than the Dirac-type equation of Eq. (2), such as, for instance the Bethe-Salpeter equation or quasipotential approaches  like the spectator equation (see, e.g. https://arxiv.org/abs/1707.09303), applied to e-h-excitations in graphene? Maybe the authors could comment on this.

In addition, I have encountered various typos (e.g. conservation low, 0ff, ranget, etc...), page 3 has too many paragraphs, the definition of the function Φ(t) on page 7 seems incorrect, the dots over E on page 9 should not be fat, and on pages 8 and 9 the English requires improvement. The last Section (8 Conclusion) is rather a summary/outline than a conclusion. This needs to be changed.

Author Response

The authors are grateful to the referee for the careful study of our manuscript and for the positive recommendation. We also thank for the insightful comments and critical remarks that helped us to improve the quality of the manuscript. Below we give point-by-point replies to the referees comments and explain the corresponding changes made in the manuscript (in red color).

1) The initial condition (after Eq. (25)) of the kinetic equation chosen in this paper is an assumption. What is the physical justification of this (if there is any?) and what would change if different initial conditions are chosen? For instance the amplitude (39) depends explicitly on the initial condition.

In the present manuscript we apply indeed initial conditions where electron and hole states are not occupied. Honestly speaking, wejust adapted these conditions from our previous studies of vacuum pair creation of an electron-positron plasma 

(dynamical Schwinger effect) in strong time-dependent external fields (e.g., Ref. [8]). We have modified the text after Eq. (25) accordingly.

One may speculate that the present case is similar to that of Mott insulators 

(see Ref. [1]) where the conduction band is empty and therefore the initial condition of absence of particles (in that band) may be reasonable.

In the case with the inertial particle production mechanism (see arxiv:1510.09196) we have already solved a KE of the type (24) with nonvanishing initial conditions, namely with a thermal initial state. The result of the Schwinger-type particle creation process (and rescattering in that work) was a nonthermal particle distribution in momentum space. Such a result we would expect also here if we were to use nonvanishing, e.g., thermal initial conditions.

We thank the referee for this valuable comment which we shall follow up in future work.

2) Various models for the electric field have been chosen and strong model dependencies are observed for the distribution function, however the carrier density is surprisingly model independent. Do the authors have an explanation for this? If so, it should be mentioned in the manuscript.

We have extended the discussion of Fig. 10 from where this observation originated,

in the text below that Figure.

3) I am a bit confused by the statement on page 18 “The corresponding perturbationtheory can be constructed both on the system of equations (27) and in the framework of the Markovian approximation (33).”I was under the impression that the Markovian approximation is essentially nonperturbative. This needs to be clarified.

The phrase “The corresponding perturbation theory can be constructed both on the system of equations (27) and in the framework of the Markovian approximation (33).” does not contain a contradiction: the KE and its Markovian approximation were both obtained by nonperturbative methods, but they can be objects of the

corresponding perturbation theory because of they are analytical at E_a = 0 (see Appendix), in contrast to the standard QED.

4) I was wondering whether the authors have considered the use of dynamical equations other than the Dirac-type equation of Eq. (2), such as, for instance the Bethe-Salpeter equation or quasipotential approaches  like the spectator equation (see, e.g. https://arxiv.org/abs/1707.09303), applied to e-h-excitations in graphene? Maybe the authors could comment on this.

We have considered before, e.g., the case of the Klein-Gordon equation for creation of boson pairs [7]. The Bethe-Salpeter equation as a starting point would be a different type of problem as it involves true interactions and thus the possibility of the formation of nonperturbative solutions such as bound and scattering states of a potential. Tackling this problem is beyond the present state of research in our work.

The reference mentioned by the referee is not solving this problem as in that work the ballistic regime is considered only when interparticle interactions are absent.

5) In addition, I have encountered various typos (e.g. conservation low, 0ff, ranget, etc...), page 3 has too many paragraphs, the definition of the function Φ(t) on page 7 seems incorrect, the dots over E on page 9 should not be fat, and on pages 8 and 9 the English requires improvement. The last Section (8 Conclusion) is rather a summary/outline than a conclusion. This needs to be changed.

We have performed a thorough Edition of the text, removing typos and improving the English language. The Conclusions have been revised too in accordance with the observation of the referee.

Round 2

Reviewer 1 Report

I am satisfied with the response.The modified Ms.

meets the criteria for acceptance. 

Reviewer 2 Report

Dear Editor,

The authors have addressed my points and revised the manuscript accordingly. The manuscript is now ready for publication.